# Detecting Impending Malnutrition of (Pre-) Frail Older Adults in Domestic Smart Home Environments

**DOI:** 10.3390/nu13061955

**Published:** 2021-06-07

**Authors:** Björn Friedrich, Jürgen M. Bauer, Andreas Hein, Rebecca Diekmann

**Affiliations:** 1Division Assistance Systems and Medical Device Technology, Department of Health Services Research, Carl von Ossietzky University, 26129 Oldenburg, Germany; bjoern.friedrich@uni-oldenburg.de (B.F.); andreas.hein@uni-oldenburg.de (A.H.); 2Center for Geriatric Medicine, Agaplesion Bethanien Krankehaus Heidelberg, University of Heidelberg, 69117 Heidelberg, Germany; juergen.bauer@bethanien-heidelberg.de

**Keywords:** geriatrics, malnutrition, older adults, smart home, preventive care

## Abstract

Malnutrition is a well-known risk factor for deteriorated physical function, disability and loss in independence in older adults. An unintended loss in body weight of more than 5% in 3 months is one indicator for malnutrition. In this study we examined the relationship between meal preparation time, hand grip strength, and body weight in order to map impending nutritional problems using ambient sensors. Data were collected in the domestic environments of 20 (pre-) frail older adults aged 85.75 y (Standard Deviation: 5.19 y) over 10-months of observation. Collecting included physical function and nutritional status of the participants and meal preparation time by a combination of motion and power sensor events. Analysis was done by rank correlation of hand grip strength, body weight, and meal preparation time. Ten participants aged 85.1 years (Standard Deviation: 4.6 y) were included. The results show a significant correlation (≥0.99) of the meal preparation time with the hand grip strength. This result validated the meal preparation time as a suitable measure for analysing the correlation between meal preparation time and body weight, and a significant correlation (≥0.99) found. Hence the meal preparation time could be used as an indicator for malnutrition. However, causalities have to be conducted by further clinical studies.

## 1. Introduction

Malnutrition as a result of insufficient nutritional intake or increased requirement is a problem of international interest and one risk factor for functional decline and loss in performance of older adults [1,2,3]. As a functional parameter the hand grip strength (HGS) is affected by malnutrition and hence the activities of daily living (ADL) as well [4,5]. One indicator for malnutrition is the significant loss in body weight. The guidelines on Enteral Nutrition: Geriatrics consider an unintentional loss of more than 5% in 3 months in body weight and a Body Mass Index (BMI) less than 20 kg/m${}^{2}$ as indicators for malnutrition [4]. A continuous monitoring of the body weight enables physicians to investigate the cause of major fluctuations of the body weight and choose a proper treatment at an early stage of an impending malnutrition. However, a continuous monitoring of the body weight by professionals exceeds their capacities and self-monitoring tends to be unreliable in the target group of older (pre-) frail adults [6]. Low-cost, unobtrusive ambient sensor systems are well-accepted among the group of older adults if installed in a flat for continuous monitoring [7,8]. The aim of this contribution is to show that the meal preparation time (key performance indicator, KPI) is associated with body weight changes as an indicator for impending malnutrition. To validate the suitability of the KPI, the relation between HGS and KPI was investigated and based on the results the relation between the KPI and the body weight was investigated. We used the data of an observational randomised controlled trial study with initially 20 participants aged 84.75 y with a Standard Deviation (SD) of 5.19 y.

This article is structured as follows. The Section 2 is a short survey of existing approaches to estimate the nutritional intake, measuring the body weight and detecting (instrumental) ADL using information technology. In Section 3 the OTAGO study for collecting the data, the preprocessing of the dataset, and the used algorithms are explained. In Section 4 the results are shown. The discussion of the results can be found in Section 5. In the last section conclusions are drawn from results and discussion.

## 2. State of the Art

This section is divided into two parts. The first part covers the state of the art of monitoring the nutritional intake and the body weight in domestic smart home environments and the second part covers the state of the art of detecting ADL in domestic smart home environments.

### 2.1. Nutritional Intake and Body Weight

A popular approach to monitor the nutritional intake in smart homes is to use smart appliances in the kitchen [9]. As part of a smart kitchen a smart fridge tracks the food and estimates the nutritional intake based on ingredients [10,11,12]. Moreover, the fridge has the capability to suggest recipes based on the estimated nutritional intake and the ingredients in the fridge.

Another essential appliance for preparing meals is the stove. When used for cooking most of the used ingredients are coming together at the stove and hence it is a suitable place for tracking the used ingredients. The quality of the nutritional intake estimation can be enhanced by communication with other smart appliances such as the smart fridge [13]. If no smart fridge is available a smartphone application for determining the nutritional facts is introduced. Adding a smart weighting scale to the already smart kitchen gives the essential information of the amount of each used ingredient and further refines the estimation [14]. An approach not relying on a smart kitchen uses groceries receipts to analyse the bought food and derive nutritional facts from that information [15].

Using smart body weighting scales is another approach to measure the body weight changes as an indicator of malnutrition. The results are quite accurate compared to in-person measurements [16]. Using ambient sensors for measuring the body weight is an interesting approach as well, because no effort by a person is needed to obtain the values. People are nearly using their beds every day and a daily weight measurement can be obtain by installing load cells under the feet of the bed [17].

The disadvantages of the smart kitchen approach are the costs and the usability. To transform a kitchen into a smart kitchen, renovation is necessary and the rehab in combination with the new appliances likely exceeds the funds of older adults. Moreover, a certain procedure for preparing meals is needed and a smartphone is involved as well. Especially, older adults are encounter difficulties using smartphones or following a certain unknown procedure.

The smart body weighing scale approach is promising, but the measurements are varying and tend to be unreliable for individuals with a body weight of more than 110 kg [16].

### 2.2. Activities of Daily Living

In Geriatrics Medicine distinction is made between activities of daily living (ADL) and instrumental activities of daily living (iADL) [18]. ADLs are daily self-care activities, like bathing, grooming and homemaking and essential for fulfilling the basic psychological and physiological needs of humans [19]. The iADLs are more complex activities of daily living, like cooking, groceries, and using means of transportation [20]. The ability to perform the (i) ADL is an indicator for independence. A common approach is to use activation sequences of sensors to define sequences and the type of the sensor, motion and power consumption, to determine the (i) ADL [21,22]. The sequences are used to train machine learning models, e.g., for prediction purposes [23]. If none or only a few ground truth labels are available unsupervised learning is used to detect activities of daily living. Based on the cluster structure all unknown samples can be labeled [24,25]. For recognition enhancement wearable sensors and smartphones are combined with the ambient sensors [26,27]. Furthermore, efforts for adding more sensors like temperature, hygrometry and microphones were made for recognising (i) ADL [28].

All these approaches have one thing in common, they are relying on a homogeneous set of sensors and mostly on labeled data. Both is not available in the dataset used for the research in this contribution. Moreover, only two approaches have been tested on real-world data of older adults [22,23]. None of the approaches are trying to find a relation between the meal preparation and the physiological parameter body weight.

## 3. Materials and Methods

### 3.1. Data Acquisition

The dataset has been collected during the OTAGO study conducted from July 2014 to December 2015 by the Carl von Ossietzky University of Oldenburg. The aim of the study was to validate the effect of the OTAGO exercise programme on (pre-)frail older adults. The participants were recruited from older adults who had previously participated in studies of the university and and from residents of a local sheltered home. The inclusion criteria was at least of all a pre-frail condition (Frailty Index = 2). In total, 20 participants (3 m, 17 f) have been included in the study and have been observed for 40 weeks with monthly measurements of a comprehensive geriatric assessment including Short Physical Performance Battery (SPPB) [29], Timed Up & Go (TUG) [30], Hand Grip Strength (HGS), Frailty Index [31] and instrumental Activities of Daily Living (iADL) [32], and the monitoring of the body weight and BMI, which have been conducted by research staff of the University of Oldenburg. The SPPB and TUG are assessing the mobility of older adults. The SPPB consists of three items, *stance, 4 m walk*, and *5-times chair rise*. All items are scored from 0 to 4 points and overall 12 points can be achieved. The higher the points, the better the mobility. The first items assesses the balance, the second item the gait speed, and the last item the lower limb strength. The TUG assess similar dimensions, but combined in one item. The persons sits on a chair, rises from the chair, walks 3 m, turns around, walks 3 m back, and sits down again. The time is measured and the faster, the time the better the mobility. The HGS of both hands were measured using a JAMAR hand-held dynamometer (JAMAR, Bolingbrook, IL). The HGS of both hands was measured twice and the highest value was used as result. The Frailty Index and iADL are measures for independence. The Frailty Index was assessed by a questionary containing the items *weight loss, exhaustion, physical activity, walk time*, and *grip strength*. Each item is scored from 0 to 1 point, where 1 point denotes impairment. A score of 2 indicates pre-frail condition and a people with a score of 3 or higher are considered as frail. The The iADL questionary contains the eight items *ability to use telephone, shopping, food preparation, housekeeping, laundry, mode of transportation, responsibility for own medications*, and *ability to handle finances*. The items are scored with 0 or 1 points, where 0 is the lowest functional level and 1 the highest. The participants did not report their weight, because self–reporting is less reliable than measurements by professionals. Participants may not pay attention to measure their weigth at the same time, wearing the same apparel. Moreover, a diary containing various information like the cognitive state of health, the recent activities, special events, and visitors was kept by the research staff.

During the study a multi-component sensor system comprised of power and home automation sensors were installed in the flats of the participants. The home automation sensors were motion sensors installed in the rooms and door sensors installed at the main doors and the fridge. The sensors were connected wirelessly to a base station. The flats of the participants had different layouts and a different set of installed sensors, because the participants were free to choose the sensors. In Appendix B the Table A1 shows the available sensors for this research for each participant. Figure 1 shows two different flats of participants. The Table 1, Table 2, Table 3 and Table 4 show the values at baseline, end, and of the subcohort considered for the analysis respectively. Figure 2 features the progress of the body weight of the used subcohort over the whole study.

We used the The Strengthening the Reporting of Observational Studies in Epidemiology (STROBE) reporting guidelines to ensure all important information were included in the article [33]. The checklist can be found in the Appendix A.

### 3.2. Data Preprocessing

Before the preprocessing on data level three exclusion criteria were applied. Two on participant level and one on data level. The first criteria was the *self supporter* criteria. All participants who were not preparing their meals themselves were excluded as well as participants who have been assisted by another person. Accordingly, 2 participants were excluded, because they received meals on wheels, 6 participants, because they were always eating out and 2 participants, because they had assistance. Two participants were partly excluded, because their condition changed from *self supporters* to being assisted. Comparing the iADL questionnaires for the excluded month and participants confirmed their status as none self supporters. The second criteria was *weight change*. If there was no weight change, the month was excluded. A change of 1% or less was considered as no change, because the measurements might have not been performed at the same time and with the same apparel. For all remaining participants a weight change was recorded and 4 month of 4 participants, 1 month for each participant, were excluded. On data level *faulty and unreliable* data has been excluded. Reasons for faulty data are malfunctioning of the sensor base station, failure of the sensors themselves or removed and rearranged sensors by the participant. Applying the criteria 5 months were excluded. After applying all exclusion criteria 10 participants with 7.2 month (SD 2.9) were considered. The group of the excluded participants is too heterogeneous to serve as control group.

The power consumption sensors were measuring the standby power consumption of the appliances and hence the standing by time must be filtered. Using the preprocessed dataset, the time spent for preparing meals was estimated each day.

### 3.3. Meal Preparation Time Estimation

For estimating the time, i.e., key performance indicator (KPI), all sensors in relation to the kitchen were used. Since the sensor setup was different in each flat, the number of sensors was used for computing the time and not the used appliance. In the left flat in Figure 1 three sensors were installed in the kitchen, one motion sensor, one door contact sensor in the fridge and one power consumption sensor at the kettle. In the right flat two motion sensors, one door contact sensor in fridge and four power consumption sensors at the microwave, kettle, coffee maker, and lamp were installed. The beginning of preparing a meal has been defined by a motion or door contact sensor event followed by events from at least half of the sensors in the kitchen. When there was no sensor event in the kitchen for 60 min the preparation was considered as finish. The time was computed for every day of the month. After computing the time values the function
(1)$$f\left(t\right)=a_{0}+a_{1}t$$
was minimised using the method of least squares
(2)$$\sum_{i=1}^{n}{\left(f\left(t_{i},a_{0},a_{1}\right)-t_{i}\right)}^{2}$$
where *n* is the number of days, $t_{i}$ the computed time for day *i* and, $a_{0}$ and $a_{1}$ the parameters of the linear function. The parameters $a_{0}$ and $a_{1}$ are adjusted according to the minimisation criteria
(3)$$\min_{a_{0},a_{1}}||f-{t||}_{2}^{2}$$

### 3.4. Minimal Clinically Important Difference

The minimal clinically important difference (MCID) is the smallest change in a parameter which is identified as important to the patient [34,35]. The considered parameter and the smallest change are not universally valid, i.e., there are different parameters and a different MCIDs. The MCID for the change in body weight in relation to malnutrition was defined in the ESPEN guidelines and is 5% in 3 months [4].

### 3.5. Statistical Analysis

For assessing the relationship the rank correlation coefficient Spearman’s ρ is used. Spearman’s ρ assess a monotonic relationships between two variables. In the following ρ(X,Y) denotes the correlation coefficient between the variables *X* and *Y*. All correlations with a *p*-value smaller than 0.001 are considered as significant. We used the *Scientific computing tools for Python (SciPy)* (v1.4.1) for the statistical analysis [36].

## 4. Results

The characteristics of the cohort and the sub-cohort used for this research are shown in Table 1, Table 2, Table 3 and Table 4. The results of our analysis’ are shown in the Table 5, Table 6, Table 7, Table 8, Table 9, Table 10, Table 11, Table 12, Table 13 and Table 14. Each table contains the results of one participant and shows the correlation between KPI and body weight (ρ(KPI,BodyWeight)), if MCID, if the participant gained or lost weight (*BW change*), the correlation between the KPI and HGS (ρ(KPI,HGS)), and if HGS increased or decreased (*HGS change*). All correlations with the body weight and between the HGS and the body weight and between the KPI and the body weight were greater or equal ρ=|0.99| and significant with p<0.001. The second to the fourth columns of the tables show the correlation between the body weight and the KPI, whether the month is part of a MCID change in body weight and the direction of the change, i.e., gain, loss, no change. The columns five and six show the correlation between the hand grip strength (HGS) and the KPI and the direction as well. The average time spent per day for preparing meals was 10.62 min (SD 19.57) with a minimum of 0.0 min and a maximum of 107.23 min. Overall, there were 8 months with a change in the body weight greater than 5% for 2 participants (ID 1, 4). In two of these months the participants gained weight, in another two the participants weight did not change, and in 4 months the participants lost weight. In 26 months the participants gained weight (ID 1–10) and in 20 months they (ID 1–10) lost weight. There are 18 months were no weight change was measured for participants 1, 2, 3, 5, 6, 7, 8, and 10 and the correlation coefficient could not be computed, i.e., not applicable (N/A).

For the HGS a decrease was measured for 28 months for all participants and an increase for 24 months for all participants except for 9. In 18 months the HGS analysis was not applicable (N/A).

The Figure 3, Figure 4, Figure 5 and Figure 6 are showing example plots of the values for 1 months. Figure 3 shows the first month of participant 1 and an increase in body weight (61.9–69.0 kg), a slight decrease in HGS (24.0–23.0 kg), and an increase in the KPI (≈10–15 min). Figure 4 shows the sixth month of participant 1. The weight gain was smaller than 1% (69.5–70.0 kg) and hence the month was considered as no change. The HGS changed slightly (22.0–21.7 kg) and the KPI was nearly constant. Figure 5 shows the second month of participant 4 with a significant loss in body weight (54.1–48.8 kg), a slight decrease in HGS (12.0–11.3 kg) and a strong increase in the KPI (≈39–76 min). Figure 6 shows the third month of participant 4 with a slight loss in body weight (48.8–47.4 kg), a slight decrease in HGS (11.3–11.0 kg), and a strong increase in the KPI (≈36–88 min). This month is still part of a three months interval where a MCID occurs.

## 5. Discussion

The results showed a strong correlation between the HGS and the KPI as well as between the KPI and the body weight. Especially, in all month where a MCID occurs our method shows strong correlations. The last three months of participant 8 (Table 12) are all considered as MCID, even though there was no change in weight in the first two month of the three months interval. The loss in body weight was higher than 5% in the last month only and hence alarmingly. Participant 4 had a MCID in all observed months and even though only two power consumption sensors and one motion sensor were available for the analysis, strong correlation were found in each month. The diary revealed that the mental condition of participant 4 was not well and became worse in the considered months. After the last considered month the participant was admitted to the hospital and after discharge the participant moved to an assisted housing scheme. The loss in body weight might be a consequence of decreasing cognitive function and mental condition, concluding causalities from these correlations would be too early. Reasons are that neither combination of positive and negative correlations is consistent, nor the magnitude is. In general, there are all possible increasing and decreasing combinations for the HGS, KPI, and body weight and also the sign of the correlation coefficient. Therefore, concluding causalities is not possible. A lot of factors may cause the change in one value. For example if the HGS is decreasing, and the KPI is decreasing as well, one reason could be the participant prepares simple meals, another reason could be that the participant eats pre-prepared meals or convenience food. Other possible explanations are the absence of ground truth label available for the iADL cooking and the heterogeneous sensor setup. The minimum set of sensors installed in a kitchen were one motion sensor, one door contact sensor in the fridge, and one power consumption sensor at the kettle. This may also resulted in the short average meal preparation time per day of 10.62 min. It is easy to construct a situation where a meal is prepared, but not two or more sensors are active. It is also easy to construct a situation where all sensors are active, but no meal is prepared. Therefore, the computed KPI is prone to measurement errors. Moreover, the body weight may have measurement inaccuracy, because it was not measured at the same time of the day and with various apparel. The diaries, the exclusion criteria *self supporter* was based on, were probably not completely precise, because the diaries were kept retrospectively and information were added based on participant’s statements. Participants were probably embarrassed to admit being assisted or uncapable of supporting themselves. Furthermore, there might also be a possibility that there are external factors which were not kept in the diaries.

Overall, there are variety of factors which can influence the change in body weight and information about the nutritional value of the prepared meals (e.g., minerals, vitamins), which were not tracked in the study, would give better insights in the nutritional status of the participants.

## 6. Conclusions

The aim of this contribution was to show that the meal preparation time (KPI) is associated with body weight changes as an indicator for malnutrition. The results showed that the KPI measured by home automation and power sensors is associated with the change in HGS and the change in body weight measured by professional staff. Using this association one indicator for impending malnutrition may could be detected at an early stage. However, it is too early to conclude causalities from the correlation. For finding whether there are causalities or not, a study designed for that specific purpose is needed. The study should at least survey the meal preparation time, the nutritional status, and the nutritional intake of the participants.

## Figures and Tables

**Figure 1 nutrients-13-01955-f001:**
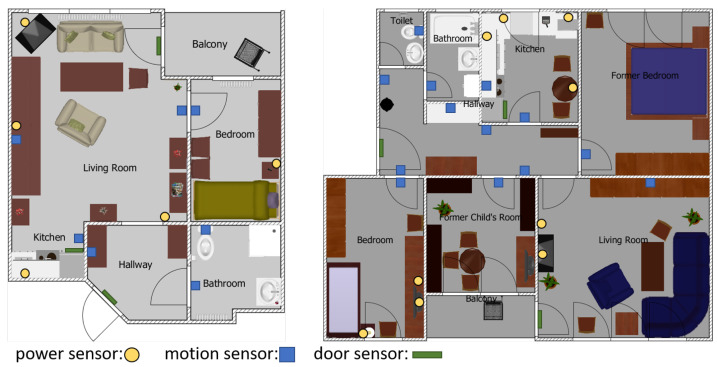
The flats of two participants of the OTAGO study. The layout and size of the flats are different and so are the installed sensors. In the right flat a great variety of sensors were installed, especially in the kitchen and the hallway.

**Figure 2 nutrients-13-01955-f002:**
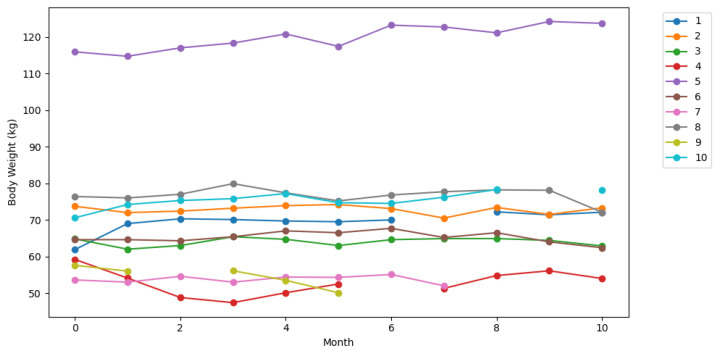
The progress of the body weight of the used subcohort over the whole study duration. Gaps in the graphs indicate missing values. The numbers in the legend are the participant IDs.

**Figure 3 nutrients-13-01955-f003:**
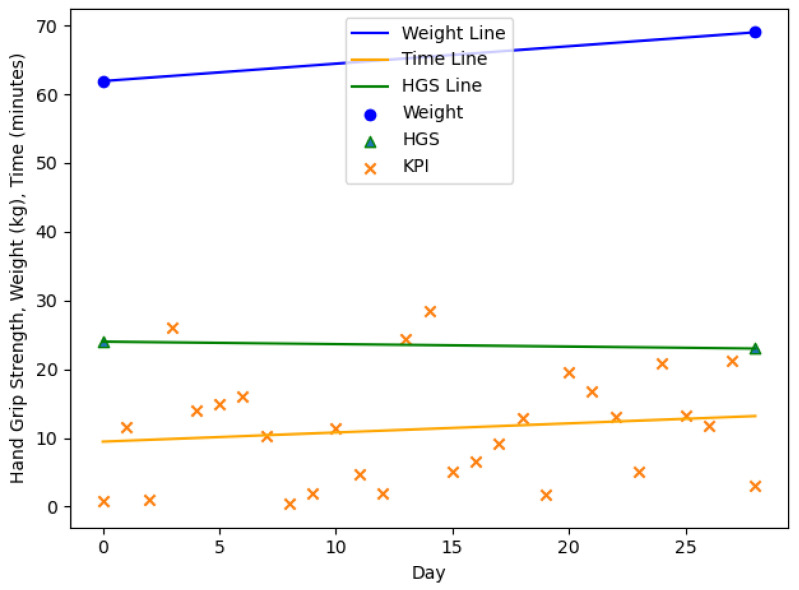
The first month of participant 1. Gain in body weight (61.9–69.0 kg), decrease in HGS (24.0–23.0 kg), increase in the KPI (≈10–15 min). HGS: Hand Grip Strength, KPI: Key Performance Indicator.

**Figure 4 nutrients-13-01955-f004:**
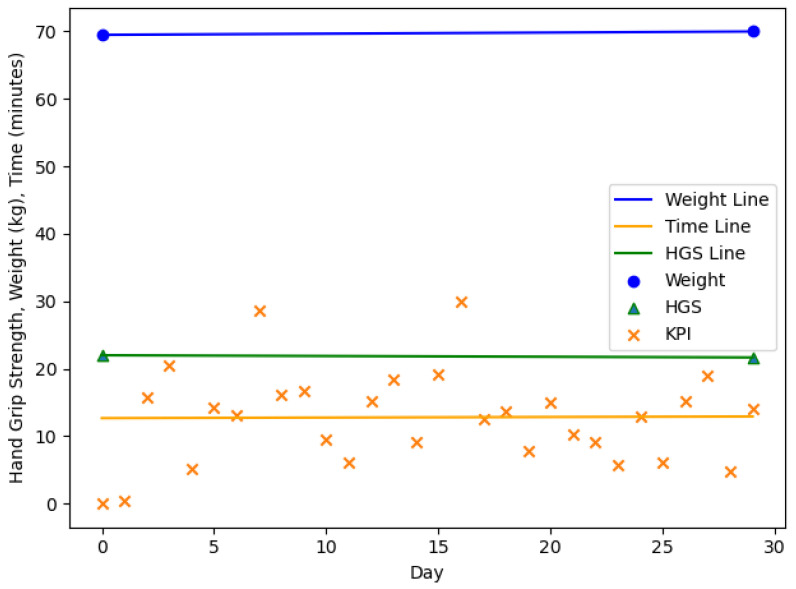
The sixth month of participant 1. Gain in body weight smaller than 1% (69.5–70.0 kg) and considered as no change, decrease in HGS (22.0–21.7 kg), nearly constant KPI (≈12–13 min). HGS: Hand Grip Strength, KPI: Key Performance Indicator.

**Figure 5 nutrients-13-01955-f005:**
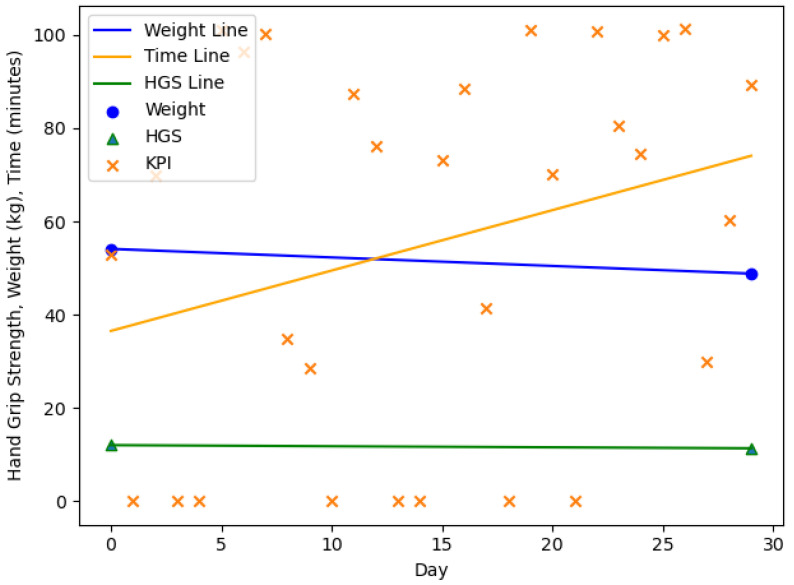
The second month of participant 4. Significant loss in body weight (54.1–48.8 kg), decrease in HGS (12.0–11.3 kg), increase in the KPI (≈39–76 min). HGS: Hand Grip Strength, KPI: Key Performance Indicator.

**Figure 6 nutrients-13-01955-f006:**
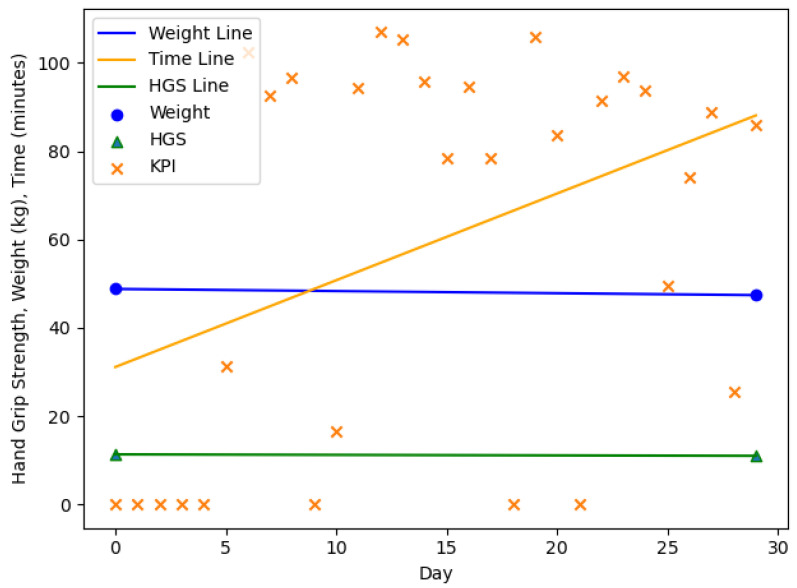
The third month of participant 4. Loss in body weight (48.8–47.4 kg), decrease in HGS (11.3–11.0 kg), increase in the KPI (≈36–88 min). HGS: Hand Grip Strength, KPI: Key Performance Indicator.

**Table 1 nutrients-13-01955-t001:** The characteristics of the study cohort at baseline (T0).

***n* = *20*, *m* = *3*, *f* = *17***	**Age (*y*)**	**Body Weight (kg) *m*/*f***	**HGS (kg) *m*/*f***	
Mean	84.8	66.2/69.2	17.1/14.1	
SD (±)	5.2	3.6/17.3	5.1/7.0	
Range (min–max)	76.0–92.0	61.9/43.8–70.8/115.9	11.7/3.7–24.0/33.0	
***n* = *20*, *m* = *3*, *f* = *17***	**Frailty Index (pts.)**	**SPPB (pts.)**	**TUG (s)**	**iADL (pts.)**
Mean	1.9	6.0	17.9	7.3
SD (±)	0.7	2.3	5.3	1.4
Range (min–max)	1.0–3.0	3.0–11.0	11.2–31.6	3.0–8.0

SD: Standard Deviation, HGS: Hand Grip Strength, SPPB: Short Physical Performance Battery, TUG: Timed Up & Go, iADL: instrumental Activities of Daily Living.

**Table 2 nutrients-13-01955-t002:** The characteristics of the study cohort at the end (T10).

***n* = *18*, *m* = *3*, *f* = *15***	**Age (*y*)**	**Body Weight (kg) *m*/*f***	**HGS (kg) *m*/*f***	
Mean	84.5	66.1/71.0	16.8/12.2	
SD (±)	4.9	5.4/19.1	4.2/3.7	
Range (min–max)	77.0–93.0	59.1/42.7–72.1/123.7	13.3/5.0–22.7/18.0	
***n* = *18*, *m* = *3*, *f* = *15***	**Frailty Index (pts.)**	**SPPB (pts.)**	**TUG (s)**	**iADL (pts.)**
Mean	2.0	6.6	16.4	6.1
SD (±)	1.0	2.9	6.0	2.3
Range (min–max)	0.0–4.0	2.0–12.0	8.5–30.06	1.0–8.0

SD: Standard Deviation, HGS: Hand Grip Strength, SPPB: Short Physical Performance Battery, TUG: Timed Up & Go, iADL: instrumental Activities of Daily Living.

**Table 3 nutrients-13-01955-t003:** The characteristics of the used subcohort at baseline (T0).

***n* = *10*, *m* = *1*, *f* = *9***	**Age (*y*)**	**Body Weight (kg) *m*/*f***	**HGS (kg) *m*/*f***	
Mean	85.1	61.9/70.7	24.0/15.4	
SD (±)	4.6	0.0/18.6	0.0/6.1	
Range (min–max)	77.0–91.0	61.9/53.6–61.9/115.9	24.0/7.3–24.0/23.7	
***n* = *10*, *m* = *1*, *f* = *9***	**Frailty Index (pts.)**	**SPPB (pts.)**	**TUG (s)**	**iADL (pts.)**
Mean	1.8	5.7	18.3	7.8
SD (±)	1.0	2.1	6.1	0.6
Range (min–max)	1.0–3.0	3.0–9.1	12.0–31.6	6.0–8.0

SD: Standard Deviation, HGS: Hand Grip Strength, SPPB: Short Physical Performance Battery, TUG: Timed Up & Go, iADL: instrumental Activities of Daily Living.

**Table 4 nutrients-13-01955-t004:** The characteristics of the used subcohort at the end (T10).

***n* = *10*, *m* = *1*, *f* = *9***	**Age (*y*)**	**Body Weight (kg) *m*/*f***	**HGS (kg) *m*/*f***	
Mean	85.1	72.1/74.8	22.7/14.5	
SD (±)	4.5	0.0/21.2	0.0/5.0	
Range (min–max)	77.0–91.0	72.1/54.0–72.1/123.7	22.7/8.0–22.7/22.7	
***n* = *10*, *m* = *1*, *f* = *9***	**Frailty Index (pts.)**	**SPPB (pts.)**	**TUG (s)**	**iADL (pts.)**
Mean	2.3	5.9	15.5	7.1
SD (±)	1.1	2.5	5.3	1.6
Range (min–max)	1.0–4.0	3.0–10.0	8.5–22.3	4.0–8.0

SD: Standard Deviation, HGS: Hand Grip Strength, SPPB: Short Physical Performance Battery, TUG: Timed Up & Go, iADL: instrumental Activities of Daily Living.

**Table 5 nutrients-13-01955-t005:** The results of the correlation analysis for participant 1.

ID	ρ(KPI,BodyWeight)	MCID	BW Change	ρ(KPI,HGS)	HGS Change
1	0.99	no	gain	−0.99	decrease
	−1.00	no	gain	1.00	decrease
	N/A	no	no change	N/A	N/A
	N/A	no	no change	N/A	N/A
	N/A	no	no change	N/A	N/A
	N/A	no	no change	N/A	N/A
	−1.00	no	gain	−1.00	increase
	1.00	no	gain	−1.00	decrease
	−1.00	no	loss	1.00	increase
	N/A	no	no change	N/A	N/A

KPI: Key Performance Indicator, MCID: Minimal Clinically Important Difference, BW: Body Weight, HGS: Hand Grip Strength, N/A: Not Applicable.

**Table 6 nutrients-13-01955-t006:** The results of the correlation analysis for participant 2.

ID	ρ(KPI,BodyWeight)	MCID	BW Change	ρ(KPI,HGS)	HGS Change
2	1.00	no	loss	1.00	decrease
	N/A	no	no change	N/A	N/A
	−1.00	no	gain	1.00	decrease
	N/A	no	no change	N/A	N/A
	N/A	no	no change	N/A	N/A
	1.00	no	loss	1.00	decrease
	−1.00	no	loss	1.00	increase
	1.00	no	gain	1.00	increase
	1.00	no	loss	1.00	decrease
	−1.00	no	gain	−1.00	increase

KPI: Key Performance Indicator, MCID: Minimal Clinically Important Difference, BW: Body Weight, HGS: Hand Grip Strength, N/A: Not Applicable.

**Table 7 nutrients-13-01955-t007:** The results of the correlation analysis for participant 3.

ID	ρ(KPI,BodyWeight)	MCID	BW Change	ρ(KPI,HGS)	HGS Change
3	−0.99	no	loss	−0.99	decrease
	−1.00	no	gain	N/A	no change
	1.00	no	gain	−1.00	decrease
	−1.00	no	loss	1.00	decrease
	1.00	no	loss	1.00	decrease
	1.00	no	gain	1.00	increase
	N/A	no	no change	N/A	N/A
	N/A	no	no change	N/A	N/A
	−1.00	no	loss	1.00	increase

KPI: Key Performance Indicator, MCID: Minimal Clinically Important Difference, BW: Body Weight, HGS: Hand Grip Strength, N/A: Not Applicable.

**Table 8 nutrients-13-01955-t008:** The results of the correlation analysis for participant 4.

ID	ρ(KPI,BodyWeight)	MCID	BW Change	ρ(KPI,HGS)	HGS Change
4	1.00	yes	loss	1.00	decrease
	−1.00	yes	loss	−1.00	decrease
	−1.00	yes	loss	−1.00	decrease
	−1.00	yes	gain	1.00	decrease
	−1.00	yes	gain	−1.00	increase

KPI: Key Performance Indicator, MCID: Minimal Clinically Important Difference, BW: Body Weight, HGS: Hand Grip Strength, N/A: Not Applicable.

**Table 9 nutrients-13-01955-t009:** The results of the correlation analysis for participant 5.

ID	ρ(KPI,BodyWeight)	MCID	BW Change	ρ(KPI,HGS)	HGS Change
5	−1.00	no	gain	−1.00	increase
	1.00	no	gain	−1.00	decrease
	N/A	no	no change	N/A	N/A
	−1.00	no	loss	1.00	increase
	−1.00	no	gain	1.00	decrease

KPI: Key Performance Indicator, MCID: Minimal Clinically Important Difference, BW: Body Weight, HGS: Hand Grip Strength, N/A: Not Applicable.

**Table 10 nutrients-13-01955-t010:** The results of the correlation analysis for participant 6.

ID	ρ(KPI,BodyWeight)	MCID	BW Change	ρ(KPI,HGS)	HGS Change
6	−0.99	no	loss	0.99	increase
	−1.00	no	gain	1.00	decrease
	−1.00	no	gain	−1.00	increase
	−1.00	no	gain	1.00	decrease
	−1.00	no	loss	1.00	increase
	−1.00	no	gain	−1.00	increase
	N/A	no	no change	N/A	N/A
	1.00	no	loss	−1.00	decrease
	-1.00	no	gain	1.00	decrease
	N/A	no	no change	N/A	N/A

KPI: Key Performance Indicator, MCID: Minimal Clinically Important Difference, BW: Body Weight, HGS: Hand Grip Strength, N/A: Not Applicable.

**Table 11 nutrients-13-01955-t011:** The results of the correlation analysis for participant 7.

ID	ρ(KPI,BodyWeight)	MCID	BW Change	ρ(KPI,HGS)	HGS Change
7	0.99	no	loss	0.99	increase
	−1.00	no	gain	−1.00	increase
	1.00	no	loss	1.00	decrease
	1.00	no	gain	1.00	increase
	N/A	no	no change	N/A	N/A
	−1.00	no	loss	1.00	increase

KPI: Key Performance Indicator, MCID: Minimal Clinically Important Difference, BW: Body Weight, HGS: Hand Grip Strength, N/A: Not Applicable.

**Table 12 nutrients-13-01955-t012:** The results of the correlation analysis for participant 8.

ID	ρ(KPI,BodyWeight)	MCID	BW Change	ρ(KPI,HGS)	HGS Change
8	N/A	no	no change	N/A	N/A
	−1.00	no	gain	−1.00	increase
	−1.00	no	gain	−1.00	increase
	−1.00	no	loss	1.00	increase
	−1.00	no	loss	−1.00	decrease
	−1.00	no	gain	−1.00	increasae
	1.00	no	gain	−1.00	decrease
	N/A	yes	no change	N/A	N/A
	N/A	yes	no change	N/A	N/A
	1.00	yes	loss	-1.00	increase

KPI: Key Performance Indicator, MCID: Minimal Clinically Important Difference, BW: Body Weight, HGS: Hand Grip Strength, N/A: Not Applicable.

**Table 13 nutrients-13-01955-t013:** The results of the correlation analysis for participant 9.

ID	ρ(KPI,BodyWeight)	MCID	BW Change	ρ(KPI,HGS)	HGS Change
9	0.99	N/A	loss	0.99	decrease

KPI: Key Performance Indicator, MCID: Minimal Clinically Important Difference, BW: Body Weight, HGS: Hand Grip Strength, N/A: Not Applicable.

**Table 14 nutrients-13-01955-t014:** The results of the correlation analysis for participant 10.

ID	ρ(KPI,BodyWeight)	MCID	BW Change	ρ(KPI,HGS)	HGS Change
10	−1.00	no	gain	1.00	decrease
	1.00	no	gain	−1.00	decrease
	N/A	no	no change	N/A	N/A
	−1.00	no	gain	−1.00	increase
	−1.00	no	loss	−1.00	decrease

KPI: Key Performance Indicator, MCID: Minimal Clinically Important Difference, BW: Body Weight, HGS: Hand Grip Strength, N/A: Not Applicable.

## Data Availability

Data is not publicly available due to privacy concerns.

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
