# Peer review of "Detecting Impending Malnutrition of (Pre-) Frail Older Adults in Domestic Smart Home Environments"

_nutrients, 2021, doi:10.3390/nu13061955_

Round 1

Reviewer 1 Report

Dear Authors, following my suggestions to improve your manuscript.

Abstract

Please adapt your content to the structure of a correctly structured abstract. At the moment, as an example, you give the information of the research design at the end of the abstract, which does not correspond to the correct order of an abstract. Also, you need to make clear in your abstract which part of your manuscript you are describing, e.g. the methods include …., the results show that ….

Please include more information about the methods, like the specific design, data collection methods, sample size of analyzed participants etc.

Author guideline of the Journal: “The abstract should be a single paragraph and should follow the style of structured abstracts, but without headings: 1) Background: Place the question addressed in a broad context and highlight the purpose of the study; 2) Methods: Describe briefly the main methods or treatments applied. Include any relevant preregistration numbers, and species and strains of any animals used. 3) Results: Summarize the article's main findings; and 4) Conclusion: Indicate the main conclusions or interpretations. The abstract should be an objective representation of the article: it must not contain results which are not presented and substantiated in the main text and should not exaggerate the main conclusions”.

Line 2 in your abstract: Please write health care professionals, like physicians and nutritionists….

State of the Art

Because you assessed IADL in your study, please include IADL beside ADL in your state of the art description.

Methods

Please include more detailed information about all methods of the study (specific design, setting, sample including recruitment, inclusion and exclusion, ethic; data procedure, …). I suggest to use a reporting guideline for the design of your study (and cite it) so that you do not forget to describe important method details.  

 3.1. Data Acquisition line 103: Which specific instrument did you use for the IADL assessment? Please cite the instrument with the respective original literature of the instrument.

3.3. Meal Preparation Time Estimation: Did you not include sensors at the stove? Because in the introduction you describe the importance of the stove! If you did not include sensors, it may be something for your limitation part?!

3.5. Statistical Analysis: You need to include the information which program you used for data analysis, like SPSS.

Results and Discussion

The average time spent per day for preparing meals was 10.62 minutes (max. 107 min). This time seems very low in general. Do you have an explanation for this in the discussion section?

Reviewer 2 Report

1. It seems that the authors use highly distal indicators for
malnutrition such as meal preparation time. If so, this needs to be
better conceptually justified why such indicators have been chosen and
why they were preferable compared to alternative, more direct indicators.

2. The sample seems rather small. Has an a priori power analysis been
conducted? Are the investigations reliable with such a small sample?

3. Crucial information such as weight may need to be manually
entered/reported by the individuals. If so, what are the biases related
to this approach?

4. In general, the advantages and challenges of such technological
approaches could be discussed in more detail.

5. What are the specific conceptual advancements beyond what is already
known from prior research?

6. The practical strengths could be better illustrated with more
detailed examples.

7. What is the ratio of costs/efforts/need compared to the expected outcome?

8. Probably a lot of ingredients are unknown. If so, the present study
may not be able to say something about specific malnutrition regarding
e.g. minerals, vitamins, etc.

Round 2

Reviewer 1 Report

Dear Authors, thank you! I have no further suggestions.

Reviewer 2 Report

I believe the manuscript has been
sufficiently improved to warrant publication in Nutrients.